# Moderating Gut Microbiome/Mitochondrial Axis in Oxazolone Induced Ulcerative Colitis: The Evolving Role of β-Glucan and/or, Aldose Reductase Inhibitor, Fidarestat

**DOI:** 10.3390/ijms24032711

**Published:** 2023-01-31

**Authors:** Omnia Safwat El-Deeb, Rasha Osama Elesawy, Amira K. Eltokhy, Hanan Alsaeid Al-Shenawy, Heba Bassiony Ghanem, Fatma H. Rizk, Ramez AE Barhoma, Rania H. Shalaby, Amal M. Abdelsattar, Shaimaa S. Mashal, Kareman Ahmed Eshra, Radwa Mahmoud El-Sharaby, Dina Adam Ali, Rowida Raafat Ibrahim

**Affiliations:** 1Medical Biochemistry Department, Faculty of Medicine, Tanta University, Tanta 31527, Egypt; 2Pharmacology Department, Faculty of Medicine, Tanta University, Tanta 31527, Egypt; 3Pathology Department, Faculty of Medicine, Tanta University, Tanta 31527, Egypt; 4Clinical Laboratory Sciences Department, College of Applied Medical Sciences, Jouf University, Sakaka 72388, Saudi Arabia; 5Physiology Department, Faculty of Medicine, Tanta University, Tanta 31527, Egypt; 6Biomedical Sciences Department, Dubai Medical College for Girls, Dubai 20170, United Arab Emirates; 7Anatomy & Embryology Department, Faculty of Medicine, Tanta University, Tanta 31527, Egypt; 8Internal Medicine Department, Faculty of Medicine, Tanta University, Tanta 31527, Egypt; 9Medical Microbiology and Immunology Department, Faculty of Medicine, Tanta University, Tanta 31527, Egypt; 10Clinical Pathology Department, Faculty of Medicine, Tanta University, Tanta 31527, Egypt

**Keywords:** mitochondria, microbiota, ꞵ-glucan, fidarestat, ulcerative colitis, SCFA

## Abstract

A mechanistic understanding of the dynamic interactions between the mitochondria and the gut microbiome is thought to offer innovative explanations for many diseases and thus provide innovative management approaches, especially in GIT-related autoimmune diseases, such as ulcerative colitis (UC). β-Glucans, important components of many nutritious diets, including oats and mushrooms, have been shown to exhibit a variety of biological anti-inflammatory and immune-modulating actions. Our research study sought to provide insight into the function of β-glucan and/or fidarestat in modifying the microbiome/mitochondrial gut axis in the treatment of UC. A total of 50 Wistar albino male rats were grouped into five groups: control, UC, β-Glucan, Fidarestat, and combined treatment groups. All the groups were tested for the presence of free fatty acid receptors 2 and 3 (FFAR-2 and -3) and mitochondrial transcription factor A (TFAM) mRNA gene expressions. The reactive oxygen species (ROS), mitochondrial membrane potential (MMP), and ATP content were found. The trimethylamine N-oxide (TMAO) and short-chain fatty acid (SCFA) levels were also examined. Nuclear factor kappa β (NF-kβ), nuclear factor (erythroid-2)-related factor 2 (Nrf2) DNA binding activity, and peroxisome proliferator-activated receptor gamma co-activator-1 (PGC-1) were identified using the ELISA method. We observed a substantial increase FFAR-2, -3, and TFAM mRNA expression after the therapy. Similar increases were seen in the ATP levels, MMP, SCFA, PGC-1, and Nrf2 DNA binding activity. The levels of ROS, TMAO, and NF-kβ, on the other hand, significantly decreased. Using β-glucan and fidarestat together had unique therapeutic benefits in treating UC by focusing on the microbiota/mitochondrial axis, opening up a new avenue for a potential treatment for such a complex, multidimensional illness.

## 1. Introduction

The connection between the microbiome and the mitochondria is receiving significant attention. Therefore, understanding how they work together mechanistically is essential if we are to provide therapeutic approaches for treating illnesses. Since gut bacteria have been demonstrated to influence important transcriptional factors, co-activators, and enzymes contributing to the biogenesis of the mitochondria, recent research investigations demonstrate a two-way communications axis between the intestinal microbiota and the mitochondria [1,2]. Furthermore, the host’s ability to produce energy, control reactive oxygen species (ROS), and reduce gut inflammation are all influenced by the microbiota of the gut and its metabolites, including certain short-chain fatty acids (SCFA) [3].

The relationship between the mitochondria and the microbiota of the gut seems to be predominantly caused by signaling through endocrine, humoral, and immunological pathways from the gastrointestinal microbiota to the mitochondria, as well as the reverse [1]. Recent research revealed that non-pathogenic gut-microbiota metabolites, such as the advantageous secondary bile acids and SCFA, may affect mitochondrial functions, such as mitochondrial energy production, biogenesis, inflammatory cascades, and redox balance, making it a plausible potential treatment [4,5].

By contrast, mitochondrial functions may change the makeup and activity of the gut microbiota because they have the ability to trigger innate immune responses [6] when cellular injury and pathogenic bacteria are detected. Epithelial cells, immunological cells, and enterochromaffin cells, which are intestine-functional-effector cells, are also influenced by mitochondria [7].

Several lines of research support the idea that the microbiome plays a crucial role in the etiology of inflammatory bowel disorders (IBD), and that any alteration in the microbiome can be attributed to the emergence of intestinal inflammation. Therefore, in mice with a genetic predisposition to colitis, the disease can be prevented by providing a germ-free environment [8]. Additionally, colitis is exacerbated by the colonization of the intestine of the mouse microbiota from persons who have an IBD by changing the immune responses, as well as by the transfer of pro-inflammatory microbiota or bacteria from sick to normal mice, which can cause inflammation [9,10].

Recently, attention has been drawn to the integration of the many functions that mitochondria play in cellular activity, as well as the contribution of mitochondrial malfunction to disease etiology. Thus, a high correlation between mitochondrial dysfunction and ulcerative colitis (UC) has been shown by recent investigations, although the exact mechanism is still unclear [11]. Furthermore, since the microbiome creates a variety of tiny compounds that have the potential to significantly affect mitochondrial homeostasis, mitochondria are the organelles that respond to microbiotic signaling in the strongest way [12].

The non-starch polysaccharides known as beta-glucans (β-glucan), which are isolated from yeasts and cereals, are categorized as physiologically active substances, known as biological response modifiers [13]. They are polysaccharides that belong to the soluble-fiber fraction. The beneficial effects of β-glucan on organisms are due to their immune-stimulating qualities; they have the capacity to attach to immune-cell receptors, activate them, and control both humoral and cell-mediated immunity. Previous investigations using the sustained Lipopolysaccharides induced enteritis model demonstrated the anti-inflammatory effect of β-glucan in the upper gastrointestinal tract [14,15]. 

Microbes in the terminal portion of the small intestine ferment glucans into SCFAs [16], which stimulate free fatty acid receptors 2 and 3 (FFAR2 and FFAR3). They provide a variety of beneficial benefits for the GIT and overall health [17]. For instance, they lower gut pH, which aids in harmful-microbe-evolution resistance [18].

The conversion of aldehydes to their associated alcohols is catalyzed by the enzyme aldose reductase (AR), which is NADPH-dependent. Sorbitol buildup, which is generated from glucose by AR in the polyol pathway, is a significant contributor to diabetic morbidity. The AR inhibitors (ARIs) work to restore normalcy to the target tissue’s increased blood and sorbitol flow via the polyol pathway. Synthetically created inhibitors have been created in vast quantities, and some of these are employed therapeutically [19]. Recent research studies documented their role in enhancing mitochondrial biogenesis and preventing mitochondrial DNA damage.

Fidarestat was proven to be safe and to have no significant, permanent adverse effects despite being a powerful AR small-molecule inhibitor with a prolonged half-life compared with other ARIs [20]. In addition to significantly enhancing mitochondrial biogenesis and reducing the damage of mitochondrial DNA, recent research showed a unique dual effect of inhibiting AR by blocking the pro-inflammatory processes which are mediated by Nuclear factor kappa B (NF-κB), and augmenting the anti-inflammatory and anti-oxidative processes which are Nuclear Factor (erythroid-2)-Related factor 2 (Nrf2)-mediated [21]. Therefore, fidarestat could be a good candidate for therapy at UC.

Aim of the study: Our research study sought to provide insight into the synergistic effect of Fidarestat when combined with β-glucan in modifying the gut mitochondrial-microbiome axis in the treatment of UC.

## 2. Results

### 2.1. Effect of Oxazolone, β-Glucan and Fidarestat on Mitochondrial Parameters

The β-glucan and fidarestat counteracted the effects of the oxazolone on the measured mitochondrial parameters in the UC group. They significantly increased the mitochondrial ATP concentration and mitochondrial transmembrane potential and significantly decreased the mitochondrial ROS levels in the treated UC groups compared to the untreated UC group. Moreover, the combined β-glucan and fidarestat treatment showed a significant improvement in all the measured mitochondrial parameters compared to the single-treatment groups (Table 1).

### 2.2. β-Glucan and Fidarestat Ameliorated Oxazolone-Induced Intestinal Oxidative Stress and Inflammation

The administration of either β-glucan or fidarestat for 28 days significantly increased the intestinal reduced glutathione (GSH) and super oxide dismutase (SOD) levels in treated groups compared to the untreated UC group. Furthermore, they significantly suppressed the intestinal NF-ĸB levels compared to the untreated UC group. The administration of β-glucan along with fidarestat in group V significantly increased the intestinal GSH and SOD levels and decreased the intestinal NF- ĸB level compared to the single-treatment groups (Table 2).

### 2.3. Effect of Oxazolone, β-Glucan and Fidarestat on Intestinal Nrf2 DNA Binding Activity and Peroxisome Proliferator-activated Receptor γ Co-activator 1α (PGC1-α) Levels

Twenty-eight days of administration of β-glucan, fidarestat, or both significantly increased the intestinal Nrf2 DNA binding activity and PGC1-α levels in the treated groups compared to the untreated UC group. However, the combined treatment of β-glucan and fidarestat significantly increased both parameters in group V compared to the single treatment in group III and group IV. In addition, the PGC1-α levels in group V did not show significant changes compared to the control group (Table 3).

### 2.4. Effect of Oxazolone, β-Glucan and Fidarestat on Microbiota-Derived Metabolites

The administration of β-glucan, fidarestat, or both along with oxazolone counteracted the oxazolone’s effects as they significantly decreased the intestinal trimethylamine N-oxide (TMAO) level while increasing the levels of fecal SCFAs in the treated groups compared to the untreated UC group. The combined treatment showed significant effects compared to the single-treatment groups (Table 4).

### 2.5. Effect of Oxazolone, β-Glucan and Fidarestat on mRNA Expressions of Colonic FFAR2, Colonic FFAR3, and Mitochondrial Transcription Factor A (TFAM)

The administration of oxazolone decreased the mRNA expressions of colonic FFAR2, colonic FFAR3, and mitochondrial TFAM in the UC group compared to the control group. However, the addition of β-glucan, fidarestat, or both for 28 days significantly increased the mRNA expressions of colonic FFAR2, colonic FFAR3, and mitochondrial TFAM in the treated groups compared to the untreated UC group (Figure 1).

### 2.6. Correlation between TMAO, SCFAs, FFAR2, FFAR3, TFAM, and Measured Variables in UC Group

A Pearson’s correlation analysis showed that the intestinal TMAO was negatively correlated with the mitochondrial ATP concentration, mitochondrial transmembrane potential, GSH, SOD, intestinal Nrf2 DNA binding activity, and intestinal PGC1-α, while it was positively correlated with the mitochondrial ROS and intestinal NF-ĸB in the UC group. On the other hand, the fecal SCFAs, colonic FFAR2, colonic FFAR3, and mitochondrial TFAM mRNA expressions were positively correlated with mitochondrial ATP concentration, mitochondrial transmembrane potential, GSH, SOD, intestinal Nrf2 DNA binding activity, and intestinal PGC1-α, while they were negatively correlated with mitochondrial ROS and intestinal NF-ĸB in the UC group (Table 5).

### 2.7. The Groups’ Histopathology and Immunohistochemistry Findings

Evaluation of the untreated group revealed that, in comparison to the normal colonic mucosa (Figure 2a), the lesion is well developed with varying degrees of severity of the damage of tissue and a high Geboes score, as shown in Figure 3a. Sections examined in β-glucan-treated group showed that the Geboes score of UC cases was decreased into score 3.1 as seen in Figure 4a, the same happens in fidarestat-treated group. (Geboes score 3.2) as seen in Figure 5a. Combined treatment with both drugs group showed that all the cases were mild UC cases (Geboes score 1.2) as seen in Figure 6a. as the cases becomes almost normal with mild increased in the inflammatory cells only with no cryptitis or gland distortion.

The sections were stained by calprotectin and S100 calcium-binding protein A12 (S100A12) to evaluate inflammation and activity in the studied cases and the control specimens. the control cases demonstrated that all the cases were completely negative for both markers (Figure 2b,c). For active UC studied cases, the score of expression of both markers were positive as seen in Figure 3b,c. both markers were detected in the intramucosal cells together with the stromal and lamina propria inflammatory cells.

Sections from the β-glucan-treated group demonstrated that both markers remained positive but with weak expression as seen in Figure 4b,c. the same was seen in the fidarestat-treated group as the expression was also weak for both markers and seen mainly focally in the mucosal cells only (Figure 5b,c).In the combined therapy group, all UC instances had negative expression of both markers as seen in Figure 6b,c.

## 3. Discussion

It was discovered that gut-microbiota signaling to mitochondria changes the epithelial-barrier function, immune-cell activation, inflammasome signaling, and mitochondrial metabolism. Furthermore, persistent intestinal inflammation is linked to both mitochondrial dysfunction and dysbiosis [22]. As a result, the possibility of precise therapeutic interventions that concentrate on microbial–mitochondrial communication offers novel ways to treat a wide range of diseases and may have significant effects on how diseases are treated in the future.

In this study, we managed to target both mitochondrial and microbiome disturbances in UC, so as to provide better management for the disease. Several studies have shown significant variations in the gastrointestinal microbiomes of IBD cases When compared to healthy controls [23,24,25]. According to these studies, the disruption of gut microbial diversity is a key factor in the pathogenesis of UC.

Unlike other microbial metabolites, which are quickly degraded and are not capable of reaching measurable plasma quantities, TMAO is entirely reliant on the microbiota and is very stable over time, producing its harmful effects [25]. Therefore, its concentration could be a good indicator of the turbulence of the gut microbiota. Interestingly, because the TMAO level in IBD is elevated and may serve as a disease existence and/or activity biomarker, TMAO levels were considerably greater in IBD patients than in healthy controls [26].

The increased TMAO levels may be linked to microbial dysbiosis and inflammatory processes in common variable human immunodeficiency disorder and may also be related to exacerbated microbial dysbiosis. This might be explained by the fact that TMAO encourages NF-κB initiation, which subsequently expands the expression of inflammatory markers, and that it may also lead to the release of a number of inflammatory cytokines [27]. These findings were consistent with our findings, which showed that the UC group had considerably higher TMAO levels than the control group.

The β-glucans are a subclass of polysaccharides found in nature that are widely present in fungi, bacteria, and cereals. They are components of the cell walls of these organisms and have a variety of other biological functions [28,29]. It has been shown that feeding colitis rats β-glucans-rich oatmeal reduced the high levels of c-reactive protein (CRP), Interleukin-12 (IL-12), and Interleukin-6 (IL-6) in the walls of the colons after only seven days [30].

In addition to its anti-inflammatory properties, it is tempting to hypothesize that oat β-glucan’s acceptability and viability are excellent for reducing levels of TMAO, a colon-derived toxin, in patients with chronic kidney disease [31]. Furthermore, it was claimed that meals enriched with soluble fiber might reduce the metabolism of trimethylamine (TMA) and TMAO by 40.6 and 62.6%, respectively. Additionally, dietary fibers influence the ecology of gut bacteria and other good bacteria [32].

The SCFAs, which originate from the fermentation of dietary fiber through the gut microbiota, are known to regulate a vast array of biological processes, including inflammation, gut motility, energy metabolism, and intestinal cellular homeostasis [33]. As a result, SCFAs and FFAR-2 activation work together to control the onset of autoimmune diseases and the metabolic syndrome in both animal and human models [34,35]. Therefore, it should come as no surprise that changes in SCFA receptor (SCFAR) signaling and function result in or encourage a number of human illnesses [33].

The higher disease-activity index, reduced colon length, and severe colonic inflammation in FFAR-2-deficient mice demonstrate the role of FFAR-2 in limiting the inflammatory responses of the intestine. These findings support the theory that FFAR-2 mediates the protective effects of SCFA in the inflammation of the intestine [34,36]. Similar investigations on inflammatory intestinal diseases in animal studies have shown the protective benefits of β-glucan supplementation when given orally and intragastrically before or after colitis is chemically induced [37]. This may be explained by β-glucans’ capacity to reduce the pro-inflammatory markers’ expression in the colon, which would improve the symptoms clinically and shield the gastrointestinal tract from leukocyte infiltration, lesions, and epithelial alterations [38,39,40].

These outcomes support our findings, which indicated that the effects of β-glucans on FFAR-2 gene expression were shown by the enhanced mRNA gene expression of FFAR-2 in the treated groups compared to the untreated groups. It is plausible that by reducing the amount of TMAO in the β-glucans-treated groups, the β-glucan was able to alleviate the dysbiosis-related turbulence in UC. On the other hand, it increased FFAR-2 mRNA gene expression, supporting the hypothesis that β-glucans stimulate SCFA synthesis, which in turn increases FFAR-2 gene expression.

Although ARIs were initially approved as antidiabetic drugs, mounting evidence suggests the future use of safe and effective ARIs in alleviating various major inflammatory pathologies [41]. A new strategy for the management of the inflammatory disorders brought on by ROS is the inhibition of AR, due to its role in reducing the glutathione-lipid and lipid-aldehydes mediation of oxidative stress (OS) signals, which leads to the activation of NF-κB [42]. 

The direct relation between elevated AR activity and OS was demonstrated in recent studies. For example, AR overexpression resulted in increased TMAO levels, lipid peroxidation, and the depletion of major nonenzymatic antioxidants, i.e., GSH levels and SOD activity [43,44]. Fidarestat was remarkably effective in arresting OS, as evidenced by the preservation of normal MDA and TMAO; this remarkable antioxidant effect was attributed to the upregulation of GSH transferase, combined with a rapid turnover of the GSH redox cycle owing to increases in both GSSG-reductase and GSH-peroxidase activities [44,45].

Recent studies have shown that the overexpression of TFAM and PGC1-α reduces the production of ROS in the mitochondria and improves mitochondrial respiratory performance [46]. Furthermore, the nuclear gene, TFAM, is crucial for boosting the replication of the mitochondrial genome, which makes TFAM a key player in the process of oxidative phosphorylation, the mechanism by which energy is produced [47]. Prior reports suggested that PGC1-α and TFAM are involved in IBD [48]. Additionally, the production of proteins from mitochondria, the loss of DNA of the mitochondria, and the halting of mitochondrial respiration are all associated with tissue damage in UC [49].

Regarding the mitochondrial modulatory effect of fidarestat, PGC1-α, a key regulator of the transcriptional pathway that controls mitochondrial biogenesis, was shown to be more highly expressed when the AR inhibitor was present, according to many studies. The expression of the transcription factor, Nuclear respiratory factor 1 (NRF-1), which affects nuclear genes encoding proteins required for mitochondrial biogenesis, in addition to mitochondrial DNA replication and transcription, is particularly enhanced by PGC1-α. The expression of TFAM is co-activated by PGC1-α and Nuclear respiratory factors (NRFs) [21].

Sonowal et al., (2021) revealed in recent research that fidarestat significantly prevented the doxorubicin-induced loss of the membrane potential of the mitochondria, along with a significant increase in the mitochondrial number. Moreover, fidarestat administration ensured a substantial rise in the mitochondrial biogenesis markers, PGC1-α and TFAM. The study concluded that fidarestat can activate mitochondrial biogenesis, ultimately culminating in the effective inhibition of doxorubicin-induced inflammatory and immune responses [50].

Importantly, AR activation results in the OS-induced phosphorylation of p53, which blocks Bcl-xL’s ability to prevent apoptosis. This causes an increase in the creation of mitochondrial transition pores, the dissipation of membrane potential (ΔΨm), and the malfunction of or damage to the mitochondria [51]. In fact, higher mitochondrial ATP content and the enhanced expression of sirtuin 1 (SIRT1), AMP-activated protein kinase (AMPK), and PGC1-α were indicators that the deletion of AR boosted mitochondrial energy metabolism and biogenesis [52]. Moreover, the inhibition of AR resulted in a significant decrease in the ROS of the mitochondria, as well as a reduction in mitochondrial permeability transition pore MPTP opening, improving the ATP content, and alleviating reductions in complex I and V activities in cardiac tissues after myocardial-ischemia-reperfusion injury [53].

The therapy with ARIs resulted in a noticeable increase in antioxidant-enzyme activity, glutathione peroxidase, SOD, and catalase, as well as the downregulation of the expression of endothelial adhesions [54]. Furthermore, it exerted a substantial protective effect against high-glucose-induced cardiomyocyte injuries by attenuating oxidative stress and mitochondrial injury [55].

## 4. Materials and Methods

### 4.1. Chemicals

Fidarestat (2S,4S)-6-fluoro-2, 3-dihydro-2′, 5′-dioxo-spiro [4 H1-benzopyran-4, 4′-imidazolidine]-carboxamide, was bought from Cayman Chemical, Ann Arbor, Michigan, USA, ≥95 purity, CAS NO: 136087-85-9), and β-glucan was bought from EUSA Colors (ASIA) Limited in Tangshan, Hebei, China [40]. Oxazolone (4-Ethoxymethylene-2-phenyl-2-oxazolin-5-one, CAS NO: 15646-46-5) as well as all chemicals were purchased from Sigma (Sigma, St. Louis, MO, USA) unless otherwise stated.

### 4.2. Animals

Animals were obtained from Tanta University’s Animal House. These were 50 Wistar albino male rats, weighing between 150 and 200 g. They were kept in normal rat cages, each holding five individuals in groups of five. They had unlimited access to food and ad libitum access to water. The g/kg diet (experimental diet) ingredients were produced using Kim et al.’s protocol [56]. For control rats, it consisted of the following foods: fat 5% (corn oil 5%), carbohydrates 65% (sucrose 50% and corn starch 15%), proteins 20.3% (DL-methionine 3% and casein 20%), fiber 5%, salt combination 3.7%, and vitamin combination 1%. Animals were acclimated to these circumstances for a week before the experiment. The Medical Faculty of Tanta University followed the Guidelines for the Care and Use of Laboratory Animals in their implementation of experimental methods and animal care (Institute of Laboratory Animal Resources, 1996), with ethical-approval code 36058/11/22.

### 4.3. Experimental Design

According to a previous designation by Boirivant et al. [57], UC induction was accomplished. In brief, the oxazolone solution was prepared by dissolving it in liquid ethanol 40% (*v*/*v*) to a constant concentration of 7.5 mg/mL, and it was then delivered once in a dosage of 1.1 mL/rat via a rubber catheter implanted 4 cm into the anal canal while the rats were under mild anesthesia. As soon as possible, the catheter was withdrawn, and the rat was placed in a vertical position for half a min to ensure that the oxazolone was disseminated uniformly throughout the colon and to the cecum.

#### The Rats Were Divided into Five Equal Groups, Each with Ten Rats

**Control group:** received 1.1 mL/rat of 40% ethanol solvent once via a rubber catheter while receiving mild ether anesthesia.**UC group:** was shown the non-treated group with UC caused by oxazolone.**β-Glucan group:** for 28 days, oxazolone-induced UC was combined with intragastric gavage administration of β-glucan at a dosage of 350 mg/kg/day (at a value of 140 mg/mL in filtered water) [58].**Fidarestat group:** received fidarestat by intragastric gavage for 28 days at a dosage of 4 mg/kg at a 1 mg/mL concentration in DMSO after oxazolone-induced UC [59].**Combined treatment group:** UC caused by oxazolone with concomitant administration of fidarestat and β-glucan at similar doses and time frames previously described.

### 4.4. Tissue Collection and Homogenate Creation

Under anesthesia with 2–5% isoflurane, all rats were sacrificed by beheading. Rat intestinal tissues were separated, weighed, and finally separated into three sections: one section was used for immunohistochemical and histopathological studies; another section was sustained in formaldehyde (10%) as a buffer; the third section comprised tissue homogenates, which were used to separate the cytosol and mitochondria using centrifugation method, as previously reported [60]. In order to isolate mitochondria, cells were gathered and combined in a buffer containing sucrose 250 mM, 0.5 mM EDTA, 3 mM HEPES-NaOH, 0.5 mM EGTA, phosphatase, and protease-inhibitor combinations (with modified pH 7.2). The mix was homogenized on ice for one minute, after which it was centrifuged at 1000 *g* for ten minutes at four degrees Celsius.

The supernatant was carefully removed in order to obtain a clean mitochondrial lysate, and at 12,000 *g* at 4 °C, the pellets were centrifuged for fifteen min. Following the removal of the residual supernatant, the pellet holding the mitochondria was re-suspended for additional use. The tissue was homogenized on ice after being scissor-chipped.

Half of the tissue homogenate was centrifuged for 10 min at 9500 *g* to pellet nuclei. The other half was spun for 5 min at 4 °C at 1000 *g* to pellet-cell fragments. The supernatant from the centrifugation was stored at −80 °C for use in subsequent biochemical-marker experiments. Centrifuging of the supernatant for 25 min at 14,000 *g* produced the mitochondrial and soluble cytosolic fractions. The Lowry technique was used to calculate the protein concentrations in the tissues [61]. Finally, the third portion of tissues was kept at −80 °C until it was needed for the gene-expression investigation.

### 4.5. Mitochondrial Parameters

#### 4.5.1. Mitochondrial ATP Concentration

Using colorimetric ATP test kits provided by Elabscience Company, (Houston, TX, USA) (cat. no. E-BC-K157-S), mitochondrial ATP levels were calculated. A semi-automatic BTS-350 Biosystems spectrophotometer was employed to determine the absorbance of each specimen at 636 nm.

#### 4.5.2. Mitochondrial Transmembrane Potential (ΔΨm) Assay

The MMP was calculated using the technique developed by Maity et al. [62]. In JC-1 assay buffer at 37 °C, the extracted mitochondria (20 g) were treated with JC-1 (300 nm) for 10 m. In a spectrofluorometer, the fluorescence of each specimen was analyzed (excitation, 490 nm; slit, 5 nm; emission, 590 nm for J-aggregate and 530 nm for Jmonomer; slit, 7.2 nm).

#### 4.5.3. Mitochondria ROS Assay

This was evaluated spectrophotometrically following the instructions provided by Cayman Chemical Company (Ann Arbor, MI, USA), for their mitochondrial ROS test kits (cat no. 701600). Two replications of the positive control were employed. In positive-control wells, Antimycin A was added at a maximal concentration of 10 uM. Complex III suppresses cytochrome bc1 and prevents the transport of electrons from the heme bH center to ubiquinone by binding a component of cytochrome bH to Antimycin A. The generation of mitochondrial ROS was also noticeably enhanced by this restriction of mitochondrial electron transport [63].

#### 4.5.4. Nrf2 DNA Binding Activity

The Nrf2–DNA interaction activity was measured in accordance with the ELISA kit manufacturer’s instructions (Nrf2 Transcription Factor Assay Kit, Abcam, Waltham, MA, USA, Cat #ab207223). The nuclear extract was made using the Nuclear/Cytosol Fractionation Kit (cat. no. K266-25, BioVision, Inc., Milpitas, CA, USA), in accordance with the manufacturer’s instructions.

#### 4.5.5. PGC1α and NF-kꞵ

The PGC1α and NF-kꞵ levels were measured in accordance with the manufacturer’s instructions using an ELISA kit from Shanghai Sunred Biological Technology.

### 4.6. Redox-Status Parameters

#### 4.6.1. GSH

Utilizing a commercial kit (Biodiagnostic, Dokki, Giza Egypt) and as previously explained by Beutler et al., tissue-reduced glutathione (GSH) was evaluated [64]. The process is dependent on 5, 5′ dithiobis-2-nitrobenzoic acid (DTNB) reduction by GSH, which causes the formation of a yellow molecule. The absorbance was determined at 405 nm and represented in mg/dl, and the reduced chromogen was directly related to the concentration of GSH.

#### 4.6.2. SOD

In order to measure the tissue superoxide dismutase (SOD) activity, a commercial colorimetric kit provided by (Biodiagnostic, Dokki, Giza Egypt) was used. This test measures the enzyme’s ability to inhibit the dye, called nitro blue tetrazolium, from reduction by phenazine methosulphate.

### 4.7. Microbiota-Derived Metabolite Detection

#### 4.7.1. The Level of Trimethylamine N-Oxide (TMAO)

Making use of an equimolar combination of disodium EDTA and ferrous sulphate, the Wekell technique [65] was used to determine the tissue TMAO level. In a semi-automatic BTS-350 Biosystems spectrophotometer, the absorbance of each sample was measured at 410 nm with the blank sample serving as a point of comparison.

#### 4.7.2. Determination of SCFA in Fecal Samples

Using a previously described procedure, fecal SCFAs were isolated and quantified [66]. A total of 500 μL of saturated NaCl liquid was added to 50 mg of dried feces, which were then left to sit at ambient temperature (25 °C) for half an hour before being homogenized for 3 min in a high-speed homogenizer. Next, 20 μL of H_2_SO_4_ (10%, *v*/*v*) was added and stirred for 30 s in a vortex. Using 800 L of anhydrous ether, all the SCFAs were carefully collected, after which they were centrifuged (4 °C, 10 min, 10,000 *g*). The supernatants’ SCFA components were then defined utilizing a system of gas chromatography (Agilent 7890B) fitted with a capillary column (30 m 0.25 mm 0.25 m) (Agilent J&W DB-WAX) and a detector of flame ionization, after the remnant trace water which was present in the supernatants was completely separated utilizing anhydrous Na_2_SO_4_.

### 4.8. Numerical Estimation of Colonic (FFAR-2 and FFAR-3 in the Colon and Mitochondrial TFAM mRNA Expression by Numerical Real-Time Reverse Transcription PCR (RT–PCR)

Total RNA was separated using the manufacturer’s directions by the Gene JET RNA Purification Kit. Revert Aid H—Reverse Transcriptase was used to backward-transcribe 5 g of total RNA into cDNA. Utilizing the cDNA as a framework, the CNTF gene’s relative expression was discovered utilizing the Applied Biosystem, Step One Plus RT-PCR system (Thermo Fisher Scientific, SA, Australia).

The primers were created using software (Primer 5.0), and they had the following sequences that were unique to rats (Table 6):

Thermo Fisher Scientific, SA, Australia (Catalog# K0221) supplied 12.5 μL of the qPCR Master Mix (2X Maxima SYBR Green/ROX), to which 2 μL of cDNA material, 1 μL of reverse primer, 1 μL of forward primer, and 8.5 μL of nuclease-free water were added to create a PCR mix of 25 μL. Thermal cycling occurred under the following circumstances: 

Initial DNA denaturation was conducted for 10 min at 95 °C. This was followed by DNA-denaturation amplification (40 cycles) for 15 s each, 30 s of annealing at 60 °C, and 30 s of extension at 72 °C. For analysis of the melting curve, the temperature was raised from 63 to 95 °C upon the completion of the previous cycle. The comparative cycle threshold (Ct) technique was used to assess the relative amounts of gene expression, which were then normalized to the housekeeping gene [67].

### 4.9. Histopathological Processing and Group Assessment

After successfully simulating the lesions for 24 h, 10% formaldehyde was used to fix the mice’s lesions, after which paraffin was used to create paraffin slices. Hematoxylin and eosin were used to verify the diagnosis and assess the damage to the tissues using the streamlined Geboes scoring method [68].

Under light microscopy, the morphology of the sections was evaluated, and the tissue-damage index (TDI) was scored, Geboes score.

### 4.10. Immunohistochemical Analysis and Assessment of the Groups Examined

Sections taken from paraffin-embedded specimens and fixed in formalin were adhered to slides carrying a positive charge and then autoclaved for 24 h at 58 °C. Sections were deparaffinized and rehydrated before being soaked in 10 mL/L of superoxide anion for 20 min before being soaked three times, for 3 min each, in PBS. Before incubation with the diluted antibodies at 4 °C for 18 h and the envision reagent at 37 °C 30 min, 10 mL/L of normal goat anti-rabbit serum was used as a pretreatment for the slices at room temperature for 20 min. Sections were stained for 8 min with 0.4 g/L DAB, 0.3 mL/L H2O2, and 30 s of hematoxylin. The outcomes were examined using a light microscope.

The PBS result was utilized as a negative control in place of the main antibody. Antibodies used were calprotectin (clone MAC387, cat. no. M0747; Dako, Glostrup, Denmark) and S100A12 (mouse monoclonal/IgG, clone MSVA-812M, Dil 1:100).

Light microscopy was used to evaluate the immunohistochemical staining, using a 10× objective lens for general inspection and a 20–40× objective lens for specific confirmation. Analyzing the degree of the staining cytoplasmic positivity allowed for a semiquantitative evaluation of the immunoreactivity. Epithelial and inflammatory cells’ immunohistochemical activity was assessed. Six high-power fields identified in the mucosal areas with the greatest expression of calprotectin were used to evaluate intramucosal calprotectin expression. Irrespective of the intensity, cytoplasmic staining was considered a positive expression. Staining was graded semi-quantitatively as negative, weak, or strong. Immunostaining of tumor cells of less than 10% was scored as negative; 10–50% was scored as weak, and 51–100% was scored strong [69,70].

### 4.11. Statistical Analysis

The data were represented using the mean and the standard deviation. Data were tested for normality using Kolmogorov–Smirnov and Shapiro-Wilk tests and all exhibited approximately normal distribution. One-way analysis of variance (ANOVA) and the post hoc LSD test were used to statistically compare the various groups. Pearson’s correlations were used to test for the associations of different variables in UC. The SPSS software (version 23.0) was used to perform statistical analysis on data (IBM Corp, Armonk, NY, USA). Any *p*-values of less than 0.05 were deemed statistically significant.

## 5. Conclusions

By using the knowledge gathered, it was shown that using β-glucan and fidarestat together had unique therapeutic benefits for treating UC by focusing on the microbiota/mitochondrial axis, opening up a new avenue for a potential treatment for such a complex, multidimensional illness.

Limitations: As a further step to support data amplification and open the door to further clinical applications, in vitro research should be carried out to extrapolate our results and to acquire better mechanism deduction. The relationship between the microbiota/mitochondrial axis and UC has to be studied further in future studies.

## Figures and Tables

**Figure 1 ijms-24-02711-f001:**
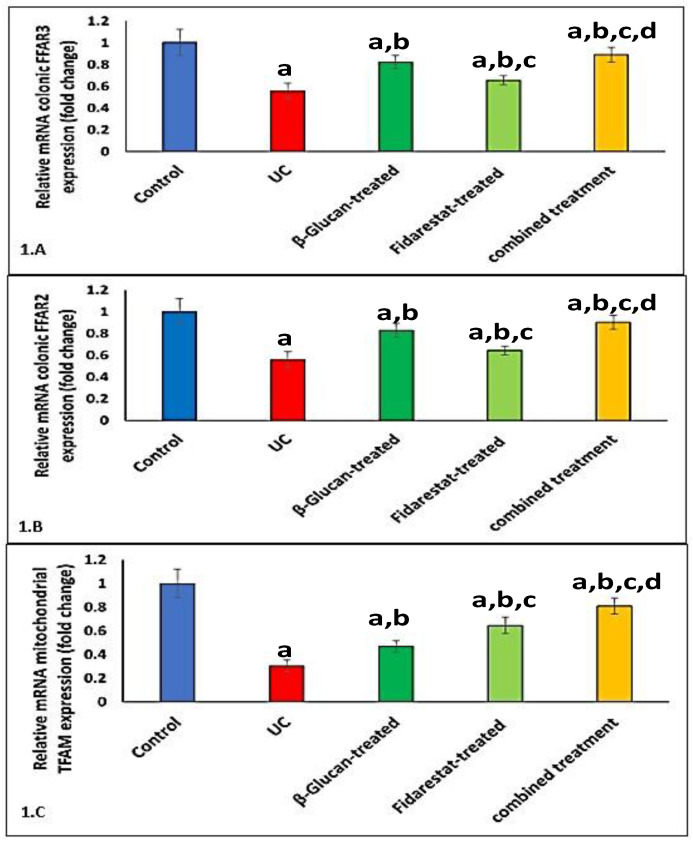
Effect of Oxazolone, β-glucan and fidarestat on relative mRNA expressions of colonic FFAR2, colonic FFAR3, and mitochondrial TFAM. (**A**). Relative mRNA expression of colonic free fatty acid receptor 2 (FFAR-2), (**B**). Relative mRNA expression of colonic free fatty acid receptor3 (FFAR3), (**C**). Relative mRNA expression of mitochondrial transcription factor A (TFAM). Data are expressed as mean ± SD. ^a^
*p* < 0.05 vs. control group, ^b^
*p* < 0.05 vs. UC group, ^c^
*p* < 0.05 vs. β-glucan-treated group and ^d^
*p* < 0.05 vs. Fidarestat-treated group.

**Figure 2 ijms-24-02711-f002:**
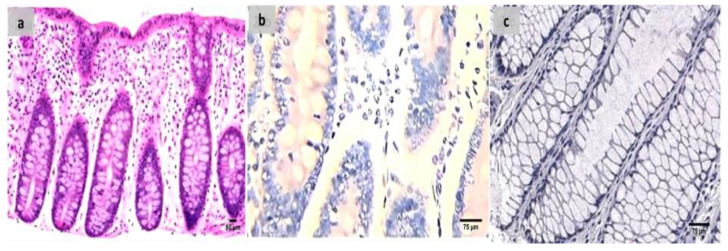
(**a**) Normal control colon specimen (H&E ×100), (**b**) normal colon stained by calprotectin showed negative expression (×200), (**c**): normal colon stained by S100A12 showed negative expression (×200).

**Figure 3 ijms-24-02711-f003:**
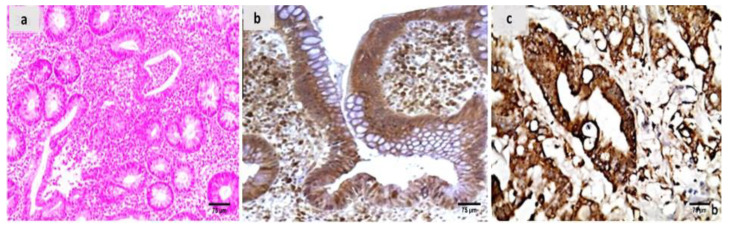
(**a**) A case of UC Geboes score 4.4 showed marked epithelial injury, cryptitis and ulcerations (H&E ×200), (**b**) the same case showed positive calprotectin expression in the mucosal cells and the stroma (×200), (**c**) also showed positive S100A12 expression in the mucosal cells (×200).

**Figure 4 ijms-24-02711-f004:**
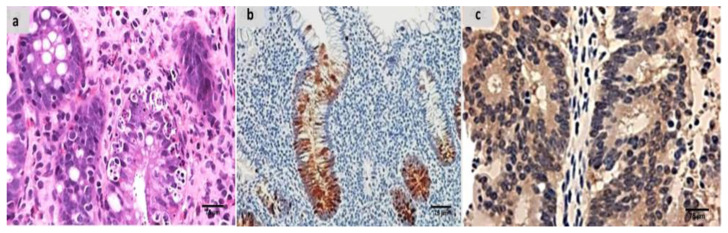
(**a**) β-glucan-treated group showing decreased activity into Geboes score 3.1 in the form of cryptitis and neutrophils seen in some of the epithelium (H&E ×200). (**b**) The same case showed less number of positive mucosal cells for calprotectin expression (×200), (**c**) also showed weak positive expression of S100A12 (×200).

**Figure 5 ijms-24-02711-f005:**
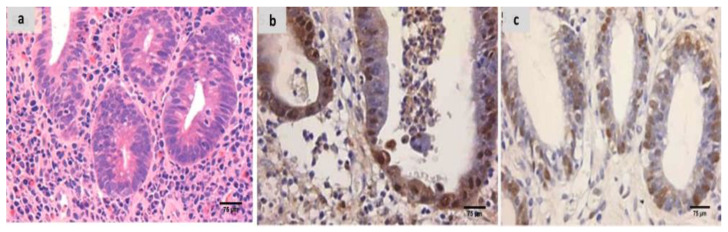
(**a**) Fidarestat treated group showing decreased activity into Geboes score of 3.2 in the form of cryptitis and neutrophils seen in most of the epithelium (H&E ×200). (**b**) The same case showed weak calprotectin but focal expression (×200), (**c**) also showed weak and focal S100A12 expression (×200).

**Figure 6 ijms-24-02711-f006:**
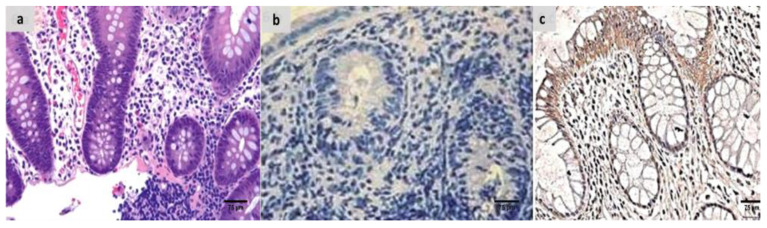
(**a**) Combined treated group showing Geboes score of 1.2 in the form of a marked increase in the basal plasma cells only with no crypt abscesses (H&E ×200). (**b**) The same case showed negative calprotectin expression (×200), (**c**) also showed negative S100A12 expression (×200).

**Table 1 ijms-24-02711-t001:** Mitochondrial parameters.

	Group I	Group II	Group III	Group IV	Group V
**ATP conc.** **(nmol/mg protein)**	13.90 ± 0.91	4.96 ± 0.94 ^a^	8.16 ± 0.72 ^a,b^	8.96 ± 0.80 ^a,b,c^	10.89 ± 0.98 ^a,b,c,d^
**Transmembrane potential (ΔΨm) (florescent unit)**	8.48 ± 0.64	2.37 ± 0.68 ^a^	4.46 ± 0.88 ^a,b^	5.63 ± 0.58 ^a,b,c^	7.82 ± 0.63 ^a,b,c,d^
**ROS** **(pmol/min/mg protein)**	5.42 ± 0.64	23.10 ± 2.18 ^a^	13.74 ± 1.24 ^a,b^	13.44 ± 1.33 ^a,b^	8.37 ± 0.93 ^a,b,c,d^

Note: Data are expressed as mean ± SD. Conc, concentration; ROS, reactive oxygen species. ^a^
*p* < 0.05 vs. group I, ^b^
*p* < 0.05 vs. group II, ^c^
*p* < 0.05 vs. group III, and ^d^
*p* < 0.05 vs. group IV. CV% of ATP conc., Transmembrane potential, ROS Assay; 3.6%, 7.5%, and 9.4% respectively.

**Table 2 ijms-24-02711-t002:** Oxidative stress and inflammatory parameters.

	Group I	Group II	Group III	Group IV	Group V
**GSH** **(μg/mg protein)**	7.99 ± 0.63	1.53 ± 0.31 ^a^	4.39 ± 0.50 ^a,b^	4.46 ± 0.55 ^a,b^	6.97 ± 0.52 ^a,b,c,d^
**SOD** **(U/gm tissue)**	405.96 ± 11.74	119.94 ± 18.45 ^a^	258.35 ± 28.71 ^a,b^	259.34 ± 32.64 ^a,b^	350.86 ± 35.06 ^a,b,c,d^
**NF-ĸB** **(ng/mg.tissue protein)**	0.18 ± 0.04	0.64 ± 0.05 ^a^	0.37 ± 0.05 ^a,b^	0.36 ± 0.07 ^a,b^	0.24 ± 0.04 ^a,b,c,d^

Note: Data are expressed as mean ± SD. GSH, reduced glutathione; SOD, super oxide dismutase; NFkB, Nuclear factor kappa B. ^a^
*p* < 0.05 vs. group I, ^b^
*p* < 0.05 vs. group II, ^c^
*p* < 0.05 vs. group III and ^d^
*p* < 0.05 vs. group IV. CV% of GSH, SOD, NF-ĸB Assay; 1.8%, 7.3%, <8% respectively.

**Table 3 ijms-24-02711-t003:** Nrf2 DNA binding activity and PGC1-α levels.

	Group I	Group II	Group III	Group IV	Group V
**Nrf2 DNA binding activity** **(μg/mg protein)**	2.06 ± 0.16	0.72 ± 0.15 ^a^	1.42 ± 0.21 ^a,b^	1.42 ± 0.17 ^a,b^	1.81 ± 0.20 ^a,b,c,d^
**PGC1-α** **(ng/mg protein)**	0.95 ± 0.11	0.30 ± 0.06 ^a^	0.56 ± 0.07 ^a,b^	0.58 ± 0.06 ^a,b^	0.88 ± 0.10 ^b,c,d^

Note: Data are expressed as mean ± SD. Nrf2, Nuclear factor (erythroid-2)- related factor 2; PGC1-α, Peroxisome proliferator-activated receptor γ co-activator 1α. ^a^
*p* < 0.05 vs. group I, ^b^
*p* < 0.05 vs. group II, ^c^
*p* < 0.05 vs. group III and ^d^
*p* < 0.05 vs. group IV. CV% of Nrf2 DNA binding activity, PGC1-α Assay; 5.5%, <10% respectively.

**Table 4 ijms-24-02711-t004:** Microbiota-derived metabolites.

	Group I	Group II	Group III	Group IV	Group V
**TMAO** **(pmol/mg protein)**	16.99 ± 3.15	97.96 ± 7.53 ^a^	65.12 ± 5.98 ^a,b^	70.63 ± 8.66 ^a,b^	35.39 ± 5.08 ^a,b,c,d^
**SCFAs (µmol/g)**	7.57 ± 0.19	1.48 ± 0.20 ^a^	4.26 ± 0.32 ^a,b^	4.33 ± 0.51 ^a,b^	6.00 ± 0.28 ^a,b,c,d^

Note: Data are expressed as mean ± SD. TMAO, trimethylamine N-oxide; SCFAs, fecal total short-chain fatty acids. ^a^
*p* < 0.05 vs. group I, ^b^
*p* < 0.05 vs. group II, ^c^
*p* < 0.05 vs. group III and ^d^
*p* < 0.05 vs. group IV. CV% of TMAO, SCFAs Assay; <10%, <12% respectively.

**Table 5 ijms-24-02711-t005:** Correlation between TMAO, SCFAs, FFAR2, FFAR3, TFAM, and measured variables in the UC group.

	TMAO		SCFAs		FFAR-2		FFAR-3		TFAM	
	*r*	*p*	*r*	*p*	*r*	*p*	*r*	*p*	*r*	*p*
**Mitochondrial ATP conc.** **(nmol/mg protein)**	−0.84 *	<0.01	0.78 *	<0.01	0.83^*^	<0.01	0.87 ^*^	<0.01	0.84 *	<0.01
**Mitochondrial transmembrane potential (ΔΨm) (florescent unit)**	−0.88 *	<0.01	0.82 *	<0.01	0.87 *	<0.01	0.90 *	<0.01	0.88 *	<0.01
**Mitochondrial ROS (pmol/min/mg protein)**	0.81 *	<0.01	−0.77 *	<0.01	−0.83 *	<0.01	−0.87 *	<0.01	−0.81 *	0.01
**GSH (μg/mg protein)**	−0.98 *	<0.01	0.95 *	<0.01	0.98 *	<0.01	0.95 *	<0.01	0.94 *	<0.01
**SOD (U/gm tissue)**	−0.80 *	<0.01	0.75 *	0.01	0.78 *	<0.01	0.83 *	<0.01	0.82 *	<0.01
**NF-ĸB** **(ng/mg.tissue protein)**	0.84 *	<0.01	−0.81 *	0.01	−0.81 *	0.01	−0.87 *	<0.01	−0.85 *	<0.01
**Nrf2 DNA binding activity** **(μg/mg protein)**	−0.91 *	<0.01	0.86 *	<0.01	0.90 *	<0.01	0.92 *	<0.01	0.90 *	<0.01
**PGC1-α** **(ng/mg protein)**	−0.84 *	<0.01	0.82 *	<0.01	0.85 *	<0.01	0.91 *	<0.01	0.88 *	<0.01

Note: UC, ulcerative colitis; TMAO, trimethylamine N-oxide; SCFAs, fecal total short chain fatty acids; FFAR-2, free fatty acid receptor 2, FFAR-3, free fatty acid receptor 3; TFAM, mitochondrial transcription factor A; Conc, concentration; ROS, reactive oxygen species. GSH, reduced glutathione; SOD, super oxide dismutase; NFkB, Nuclear factor kappa B; Nrf2, Nuclear factor (erythroid-2)- related factor 2; PGC1-α, Peroxisome proliferator-activated receptor γ co-activator 1α. * *p* < 0.05 was considered statistically significant.

**Table 6 ijms-24-02711-t006:** Primers of FFAR2, FFAR3, TFAM & GADPH.

	Forward	Reverse	Accession Number
**FFAR2**	5′CTACGAGAACTT CACCCAAGAG 3	5′ GAAGCGCCAATAA CAGAAGATG 3′	NM_001005877.2
**FFAR3**	5′ CCGGCGCAAGAGGATAAT-3′	5′ CCCACCACATGGG ACATATT 3′	NM_001108912.1
**TFAM**	5′AAGGGAATGGGAAAGGTAGA3	5′AACAGGACATGGAAAGCAGAT3′	NM_0011045
**GADPH**	5′AGACAGCCGCATCTTCTTGT 3′	5′ CTTGCCGTGGGTAGAGTCAT 3′	NM_017008.4

## Data Availability

Data will be provided upon request.

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
