# Peer review of "Moderating Gut Microbiome/Mitochondrial Axis in Oxazolone Induced Ulcerative Colitis: The Evolving Role of β-Glucan and/or, Aldose Reductase Inhibitor, Fidarestat"

_ijms, 2023, doi:10.3390/ijms24032711_

Round 1

Reviewer 1 Report

Manuscript Number: 2120994

Title:

Please make it shorter.           

This section of title is confusing. Please revise. Make it short and concise – ‘The Evolving Role of β-Glucan 3 and/or, Aldose reductase inhibitor, Fidarestat’

Abstract:

Need to add the role of beta glucan for the study.

Introduction:

Line 75 – Beta glucan is not polysaccharides, but it is a type of non-starch polysaccharides. Please amend.

Line 76: Please add reference to this statement.

Line 83: Typo.

Line 102 Make it as full sentence for the study aim. This study is study the synergistic or additional effect of Fidarestat when combined with beta glucan? Please make it clear here.

Material and Method:

Line 112: What kind of beta glucan used? From mushroom or barley? Purity of beta glucan?

Line 113: We bought more chemicals…please revise this sentence.

Line 114 & 115: …quality of all compounds quite excellent…Why need to mention this?

Line 117: Please revise the sentence: We get this animal…

Line 136: Why used 15 rats per group? What is the basis of thin number?

Line 142: What is the basis of beta glucan dosage? 350 mg/kg/day?

Line 151: Rats were sacrificed by being beheaded. How this was done?

Results:

Animal groupings: It is better to label the group name according to control/treatments Eg. Control, Fidarestat etc. Easier to follow this way.

What was the performance of each assay? Eg sensitivity. Please include under the table, ie in the caption. Please also calculate coefficient of variation for each assay. The standard deviation looks too perfect for me.

Were all data normally distributed? Please add this. Need to check the normality first before doing the statistical analyses.

Keep all values to 2 decimals.

Figures: Please improve the resolution for all figures.

Discussion:

Line 420: Please avoid the word ‘don’t’ in scientific writing.

Conclusion:

Ok. Reflecting the content.

Reviewer 2 Report

Mitochondria and gut microbiome are thought to expedite innovative explanation of many diseases and thus can provide innovative managements especially in GIT related autoimmune disease as ulcerative colitis (UC). This research study sought to provide insight on the function of β-Glucan and/or Fidarestat in modifying the microbiome-mitochondrial gut axis in the treatment of UC.

It is interesting work with good design and result. However, there still have some issues need to be revised.

1.Reactive oxygen species (ROS), mitochondrial membrane potential (MMP), and ATP content were found. Trimethylamine N-oxide (TMAO) and short chain fatty acids (SCFAs) levels were also examined. Nuclear factor kappa β (NF-kβ), nuclear factor (erythroid-2)-related factor 2 (Nrf2) DNA binding activity, and peroxisome proliferator-activated receptor gamma co-activator-1 (PGC-1) were identified. A substantial increase of FFAR-2, -3, and TFAM mRNA expression after therapy. Similar increases were seen in ATP levels, PGC-1, and Nrf2 DNA binding activity. However, the SOD, GSH-PX level also need to be tested(Oat phenolic compounds regulate metabolic syndrome in high fat diet-fed mice via gut microbiota. Food Bioscience. 50(2022)101946. Doi: 10.1016/j.fbio.2022.101946 ).

2.Indeed, there is a positive correlation of antioxidative stress and prebiotic effect, please refer this reference (The positive correlation of antioxidant activity and prebiotic effect about oat phenolic compounds. Food Chemistry, 402(2023): 134231.)

3.The levels of ROS, TMAO, and NF-kβ, on the other hand, significantly decreased when MMP and SCFA levels increased. β-glucan and Fidarestat together had unique therapeutic benefits in treating UC by focusing on the microbiota/mitochondrial axis, opening a new avenue for a potential treatment for such a complex, multidimensional illness. Please reference this reference for prebiotic effect and antioxidant activity(Whole grain benefit: oat β-glucan and phenolic compounds synergistically regulates hyperlipidemia via gut microbiota in high-fat-diet mice. Food & Function, 2022, Doi: 10.1039/d2fo01746f.)

4.How does the β-glucan deliver or packaging?( Recent advances of stimuli-responsive polysaccharide hydrogels in delivery systems: A review. Journal of Agricultural and Food Chemistry. 70(21):6300-6316.) The dosage form needs to be cleared.

5.The gut microbiota needs to be tested, since it is evaluation the microbiome-mitochondrial gut axis in IBD case (Recent advances of cereal beta-glucan on immunity with gut microbiota regulation functions and its intelligent gelling application. Critical Reviews in Food Science and Nutrition. doi: 10.1080/10408398.2021.1995842.).

Round 2

Reviewer 1 Report

Manuscript Number: 2120994

Materials and Methods:

Line 385: Please add beta glucan purity here. Beta glucan purity of 97% is quite high. Do you have the certificate of analysis (COA) for this? You can ask from supplier if you do not have one. This is very important for others to replicate the findings.

Line 388: Please change sigma to Sigma.

Line 411: Do you mean ‘mistake’ of writing 15 rats instead of 10 animals? The mistake appears twice in the Abstract and Method sections? Or do you mean mistake in the experimental design? Please

Line 412: Please remove the word ‘the’ in the animal groupings

Line 416: Simply follow the dose from previous study was not a good justification. Please improve this.

CV values for some parameters are still missing. Please add the info.

Reviewer 2 Report

The Mechanistic understanding of the dynamic interactions between mitochondria and gut microbiome is thought to expedite innovative explanation of many diseases and thus can provide innovative managements especially in GIT related autoimmune disease as ulcerative colitis (UC).

 This research study provide insight on the function of β-Glucan and/or Fidarestat in modifying the microbiome-mitochondrial gut axis in the treatment of UC. Reactive oxygen species (ROS), mitochondrial membrane potential (MMP), ATP content, Trimethylamine N-oxide (TMAO) and short chain fatty acids (SCFAs) levels were examined. Nuclear factor kappa β (NF-kβ), nuclear factor (erythroid-2)-related factor 2 (Nrf2) DNA binding activity, and peroxisome proliferator-activated receptor gamma co-activator-1 (PGC-1) were identified using the ELISA method. The levels of ROS, TMAO, and NF-kβ, on the other hand, significantly decreased when MMP and SCFA levels increased.

Using β-glucan and Fidarestat together had unique therapeutic benefits in treating UC by focusing on the microbiota/mitochondrial axis, opening a new avenue for a potential treatment for such a complex, multidimensional illness.

It is well design and the result is interesting. The manuscript has been revised and the quality has been improved. However, there still have minor revision need to improve.

1.     The sentence should be checked in the manuscript.

2.     Some refence should be supplement completed (Whole grain benefit: oat β-glucan and phenolic compounds synergistically regulates hyperlipidemia via gut microbiota in high-fat-diet mice. Food & Function, 2022, 13, 12686-12696. Doi: 10.1039/d2fo01746f.). The reference should be update in recently years.

3.     It is better to show the ROS assay with oxidative stress (Consumption of the fish oil high-fat diet uncouples obesity and mammary tumor growth through induction of reactive oxygen species in pro-tumor macrophages. Cancer Research, 2020, 80(12): 2564-2574.).
